SCIENCE FORUM

# The critical importance of vouchers in genomics

**Abstract** A voucher is a permanently preserved specimen that is maintained in an accessible collection. In genomics, vouchers serve as the physical evidence for the taxonomic identification of genome assemblies. Unfortunately, the vast majority of vertebrate genomes stored in the GenBank database do not refer to voucher specimens. Here, we urge researchers generating new genome assemblies to deposit voucher specimens in accessible, permanent research collections, and to link these vouchers to publications, public databases, and repositories. We also encourage scientists to deposit voucher specimens in order to recognize the work of local field biologists and promote a diverse and inclusive knowledge base, and we recommend best practices for voucher deposition to prevent taxonomic errors and ensure reproducibility and legality in genetic studies.

**JANET C BUCKNER\*, ROBERT C SANDERS, BRANT C FAIRCLOTH AND PROSANTA CHAKRABARTY\***

## Introduction

The genomics era has produced genome assemblies for many species. For example, GenBank – a database maintained by the National Center for Biotechnology Information (NCBI) in the US – contains over 17,000 genome assemblies from eukaryotes. However, genomics has a serious problem: studies that sequence and assemble genomes should designate a voucher – a permanently preserved specimen in a collection that is accessible to other researchers (*Leray et al., 2019*; *Pleijel et al., 2008*) – but only a minority of genomics studies have done so.

Voucher specimens are typically identified to species, labeled, catalogued, and housed in natural history museums, herbariums, or other collections of permanently preserved organisms (where they are also available to be loaned and studied) (*Peterson et al., 2007*). These research collections follow standardized archival protocols, and collections staff are charged with maintaining taxonomic information, permits, and other data associated with each specimen (*Lendemer et al., 2020*). Because the source materials for genome sequencing projects generally come from a single individual (or sometimes multiple pooled individuals that represent a single taxon), the specimen vouchering process is an indispensable first step to ensure the legal collection of accurate biological data and the replicability of genetic studies. Unfortunately, references to specimen vouchers and their associated data are frequently omitted from publications and repositories (*Figure 1*).

Although there are several important components of the vouchering process, taxonomic identification of voucher specimens is critical because proper identification is required to understand and contextualize all aspects of biology pertinent to a species (*Colella et al., 2021*). Taxonomy in most biological disciplines is based on morphological and genetic divergence (*Schoch et al., 2020*), and joint archiving of both data types is essential to verifying the identity of biological materials now and in the future. Furthermore, taxonomic revisions are often the rule rather than the exception, underscoring the importance of linking genome sequencing data and assemblies to a voucher that can be taxonomically identified, revisited, and updated, if necessary.

The lack of vouchers associated with the sequencing and assembly of genomes is problematic for the following reasons:

**\*For correspondence:**
jbuckner1@lsu.edu (JCB);
prosanta@lsu.edu (PC)

**Competing interests:** The authors declare that no competing interests exist.

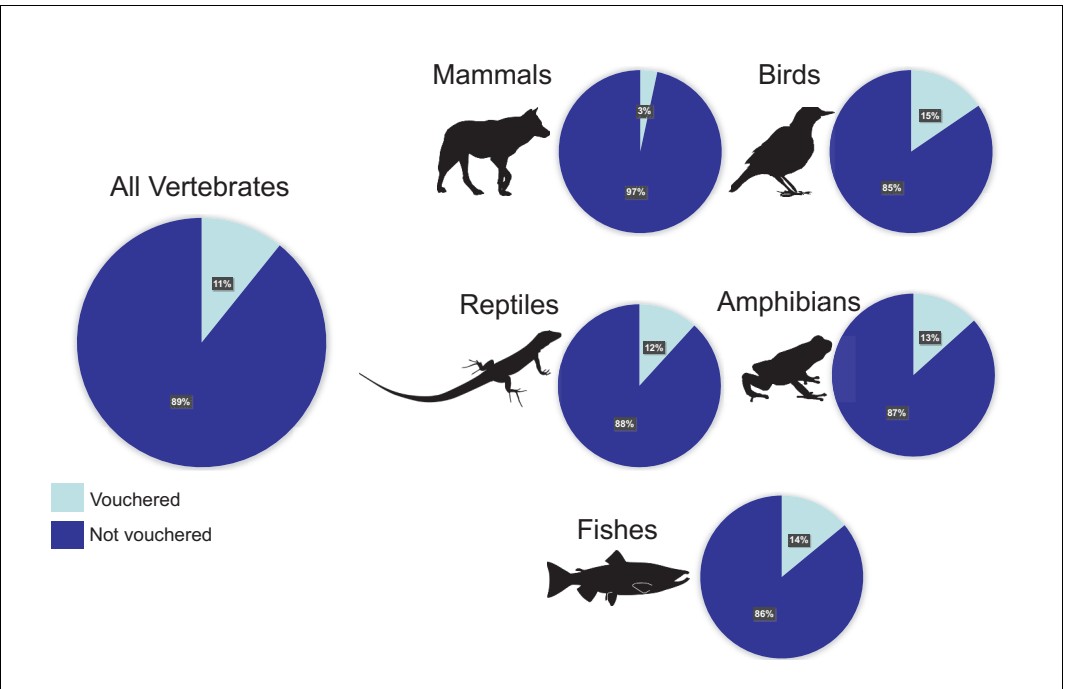

**Figure 1.** Percentages of vertebrate genomes with and without a voucher reference. Of the 1300 representative genome assemblies from vertebrate taxa that were available on GenBank (with sequence coverage greater than 30X) as of January 2020, only 11% referenced a voucher specimen in a published paper or the appropriate NCBI database field(s). The percentages for the major taxonomic groups vary from 3% of assemblies referencing a voucher for mammals to 15% of assemblies referencing a voucher for birds.

i. Genome sequencing data and genome assemblies are often assumed to be correctly identified to species; however, without a representative voucher specimen, there is only sequence-based evidence to support taxonomic identification.

ii. Some species with associated genome assemblies have undergone taxonomic revisions subsequent to sequencing, and it may be infeasible or impossible to know which species the original genomic data represent without a voucher, hindering repeatability.

iii. Future studies may propagate errors when relying on representative genomes which may have been given incorrect taxonomic assignments.

iv. Catalogued and curated biological samples (with their permit and other documentation) provide the best evidence of legal collection.

v. Local field scientists may be excluded from the scientific process when sampling/collection information is missing from repositories and publications, making genomics less inclusive.

The failure to associate voucher information with genome assemblies can lead to many real-world problems, such as slowing our understanding of emerging diseases (e.g., identifying the animal host of SARS-CoV-2 [*Thompson et al., 2021*]) to complicating clinical analyses because of the use of misidentified species (*Beaz-Hidalgo et al., 2015*).

## Limitations and the need for verifiable genomics

The best way to ensure proper taxonomic identification is through the examination of a physical voucher specimen (*Chakrabarty, 2010*; *Chakrabarty et al., 2013*; *Monckton et al., 2020*). However, there are cases when such collections and preservations are not possible. For instance, an organism may be too large to be collected and stored, too rare to be legally obtained, or so small that most of the specimen is depleted while obtaining sufficient tissue to enable sequencing and assembly. In these cases, detailed photographs should be taken to aid future identification attempts, although it should also be recognized that photographs have limited utility for taxonomy (*Monckton et al.,*

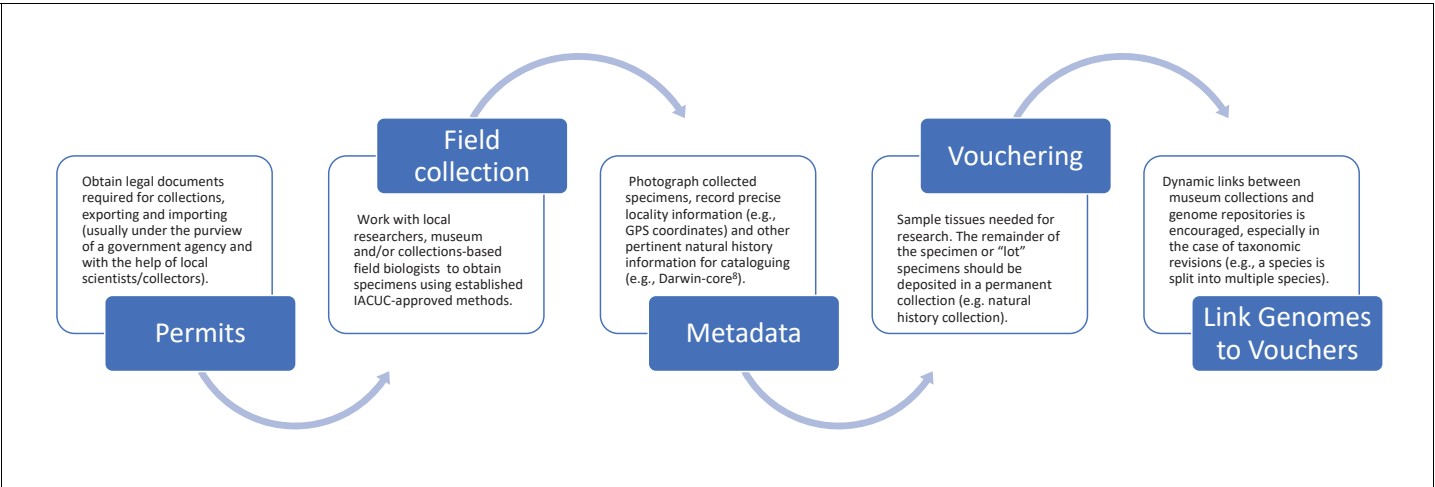

**Figure 2.** Suggested best practices for voucher-enabled genomics. Best practice starts with obtaining the necessary legal documents (see 'Permits'), and continues through fieldwork with local researchers ('Field collection'), photographing specimens and recording collection information ('Metadata'), depositing specimens ('Vouchering'), and creating dynamic links between museum collections and genome repositories ('Link Genomes to Vouchers').

*2020*; *Ceríaco et al., 2016*). Many species are distinguished on the basis of inconspicuous characters or internal anatomy that photographs might not capture. Alternatively, if other specimens exist from the same 'lot' (additional representatives of the same species from the same location and collection event), these could be treated as 'proxy' specimens for the voucher and used for future taxonomic verification (this approach is equivalent to the paragenophore voucher classification suggested in *Pleijel et al., 2008*).

One example of how designating a proxy specimen could have been helpful is in the case of the electric eel (*Electrophorus electricus*) genome assembly. After this taxon was sequenced and assembled (*Gallant et al., 2014*), a subsequent publication split this species into three, each identified by discrete phenotypic characters corresponding to different physiographic regions (*de Santana et al., 2019*). Although several eel specimens were purchased from the same aquarium vendor for tissue harvesting and nucleic acid extraction, no vouchers were saved. If additional specimens were available from the same vendor (even if not used in sequencing but assuming they were from the same locality), these could stand in proxy for the original vouchers to aid future identifications.

Similarly, individuals from the same culture/cell line/germ line/strain can be treated as proxies to aid identifications in cases where specimens used for genomic sampling are obtained from facilities maintaining these closely related individuals. Likewise, DNA samples are frequently taken from captive organisms, such as those housed in zoos and aquariums. Live organisms can be treated as vouchers and can be provided museum catalog numbers to ensure future preservation upon their death; even if a specimen is heavily dissected from a necropsy, many permanent collections are willing to preserve partial remains as vouchers.

Samples collected from organisms that are extremely large (such as blood/tissue samples taken from a whale) can also be curated and stored with other biological sample data in most natural history collections. These types of accessory or partial biological samples and photographs (or other so-called *e*-vouchers [*Monk and Baker, 2001*]) fall in the category of secondary vouchers (*Kageyama et al., 2007*) that should be used in support of vouchering whole specimens, not as alternatives – unless collecting a specimen is not possible. This holistic approach to vouchering, where primary and secondary voucher materials are collected and stored together will further increase the repeatability and reliability of genomic studies.

Theoretically, in the absence of vouchers, new specimens can be collected and molecular data from other members of a population can be used to confirm taxonomy. However, collection of additional specimens from the same location as the original may be infeasible. For example, permits to collect additional individuals may not be approved, or populations may be extirpated or replaced by closely related species before new collections can be made.

Some researchers may also argue that using organellar DNA data (e.g., mitochondrial DNA, including DNA barcoding genes) collected during the genome sequencing process will always be available as a method of taxonomic verification. However, introgression or hybridization among related species can obfuscate post-hoc taxonomic identification using molecular data, muddling the link between a voucherless-genome and subsequent genetic detective work (*Zhang and Hewitt, 1996*).

Alternatively, comparative organellar DNA can also be misidentified or unavailable from public databases such as the Barcode of Life and GenBank (*Pentinsaari et al., 2020*). For example, since the publication of the ocean sunfish (*Mola mola*) genome (*Pan et al., 2016*), the originally described taxon has been split into three distinct species (*Nyegaard et al., 2018*) with no photo or voucher from the original source animal and with the novel taxa having very limited sequence data available. In such cases, where comparative sequence data are unavailable from all recognized species of a recently split taxon, it will not be entirely clear to which species the previously sequenced genome should be assigned.

## Improving legality, equity, and inclusion in genomics

Where possible, having a proper voucher can be evidence that collections of rare or endangered species were made legally (*Colella et al., 2021*). Data associated with vouchers typically includes links to permits, field notes, and other associated documentation; without a specimen these documents are often lost because they are not associated with museums or other long-term archival research collections (*Simmons, 2017*).

Preserving representative vouchers can also make genomics more inclusive for individuals who facilitate the collection of these source materials. For example, a recent call to sequence all eukaryotic genomes (*Lewin et al., 2018*) will require the help of many in biodiversity rich but economically poor areas. These collectors of biological samples will facilitate the initiation of genome studies by obtaining local permits and source specimens, and these collectors are often the first to perform taxonomic identifications because they have first-hand knowledge of local biodiversity.

Although collection, preservation, and maintenance of domestic and international specimens should be treated as a partnership between the

scientists involved, specimen collectors are sometimes excluded from subsequent stages of the scientific process. Vouchering of specimens can serve as one mechanism among many to include collectors in the scientific process and validate their position as manuscript co-authors; the vouchering process is the first step formalizing the link between the collector and the samples critical to subsequent genomic research. Minimally, vouchering ensures the record of the collectors who enable these studies is preserved (the names of original collectors are linked to the specimens and should be perpetuated with the data obtained from their vouchers).

Support (financial as well as academic credit) for museums and the preparators who maintain these research collections and update taxonomy and reference catalogs should also not be overlooked (*Bradley et al., 2014*). Using vouchers establishes one link between the collectors, curators, collections managers, and the subsequent genomic resources – an important step for making genomics more inclusive, sharing credit for resources more equally, attracting and training participants from historically marginalized groups, and expanding the scientific infrastructure globally. Vouchering also enables a wide spectrum of scientific uses beyond genomics including additional studies of natural history and ecology and the use of specimen resources for outreach activities (*Peterson et al., 2007*; *Cook et al., 2017*).

## Suggested best practices of specimen vouchering for genomic studies

*Figure 2* outlines the process for collecting samples for preservation in natural history collections and the mechanisms for establishing proper taxonomic identification while ensuring scientific reproducibility in genomic studies. Materials taken from live organisms (in, for example, zoos or breeding facilities) should follow similar steps (see above). We encourage genetic databases and journal publishers to consider requesting these best practices as part of their submission process. We further recommend that authors include photographs of the voucher specimens in their publications describing new genome assemblies to add additional safeguards for future identification. As we enter a future when genomic analyses will be the most frequent method of genetic study, we need to avoid a scenario where it will become increasingly intractable to correctly assign species to

available genome assemblies; having a voucher specimen representing the reference genome for every species is the best solution to that increasingly difficult problem.

## Materials and methods

We surveyed the NCBI list of vertebrate genomes (focusing on reference/representative genomes of each species) with an assembly publication date up to January 1, 2020 (https://www.ncbi.nlm.nih.gov/genome/browse#!/eukaryotes/vertebrates) and coverage of 30X or greater. Although we focused on reviewing vertebrate genomes, the lack of vouchers is a problem among genetic sequences submitted from many different types of organisms (*Leray et al., 2019*; *Pleijel et al., 2008*; *Peterson et al., 2007*; *Lendemer et al., 2020*; *Colella et al., 2021*; *Schoch et al., 2020*; *Thompson et al., 2021*; *Beaz-Hidalgo et al., 2015*; *Chakrabarty, 2010*; *Chakrabarty et al., 2013*). When available, we also cross-checked the original publications reporting genome assemblies for references to a deposited voucher specimen. Sometimes, we could not find any papers associated with the genome or failed to find contact information in the NCBI. Summarized information on the genomes included in this assessment are available at: https://doi.org/10.5061/dryad.6wwpzgmz4.

### Acknowledgements

John Sullivan and Stacy Ciufo provided valuable insights into genetic databases such as NCBI. We thank all the researchers who replied with voucher and/or genomic sequence information.

**Janet C Buckner** is in the Museum of Natural Science, Louisiana State University, Baton Rouge, United States
jbuckner1@lsu.edu
https://orcid.org/0000-0001-7509-8370

**Robert C Sanders** is in the Museum of Natural Science, Louisiana State University, Baton Rouge, United States

**Brant C Faircloth** is in the Museum of Natural Science and the Department of Biological Sciences, Louisiana State University, Baton Rouge, United States
https://orcid.org/0000-0002-1943-0217

**Prosanta Chakrabarty** is in the Museum of Natural Science and Department of Biological Sciences, Louisiana State University, Baton Rouge; Carleton University and the Canadian Museum of Nature, Ottawa, Canada; the American Museum of Natural History, New York; and the National Museum of Natural History, Smithsonian Institution, Washington
prosanta@lsu.edu

https://orcid.org/0000-0003-0565-0312

*Author contributions:* Janet C Buckner, Data curation, Formal analysis, Investigation, Methodology, Writing - review and editing; Robert C Sanders, Data curation, Formal analysis; Brant C Faircloth, Writing - review and editing; Prosanta Chakrabarty, Conceptualization, Data curation, Formal analysis, Investigation, Writing - original draft, Writing - review and editing

*Competing interests:* The authors declare that no competing interests exist.

### Funding

| Funder | Grant reference number | Author |
|---|---|---|
| National Science Foundation | IOB-1754417 | Brant C Faircloth |

The funders had no role in study design, data collection and interpretation, or the decision to submit the work for publication.

### Decision letter and Author response

Decision letter https://doi.org/10.7554/eLife.68264.sa1
Author response https://doi.org/10.7554/eLife.68264.sa2

## Additional files

### Supplementary files

• Transparent reporting form

### Data availability

Summarized information on the genomes included in this assessment are available at: https://doi.org/10.5061/dryad.6wwpzgmz4.

The following dataset was generated:

| Author(s) | Year | Dataset URL | Database and Identifier |
|---|---|---|---|
| Chakrabarty P, Buckner JC, Faircloth BC | 2021 | https://doi.org/10.5061/dryad.6wwpzgmz4 | Dryad Digital Repository, 10.5061/dryad.6wwpzgmz4 |

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
