## [Decision Letter]

Thank you for submitting your manuscript "Vouchers are critical (but often overlooked) in studies of genome biology" to *eLife* for consideration as a Feature Article. Your article has been reviewed by three peer reviewers, and their comments have been combined to produce this decision letter. On the basis of the comments from the reviewers, we invite you to submit a revised version of your manuscript that addresses the points below.

Summary:

Buckner et al. surveyed the available vertebrate genomes on GenBank and discovered that most of them had no "voucher" specimen reference noted in the record or associated publications. In fact, only 11% of the GenBank genomes referenced a voucher specimen (varying from a high in birds of 15% to a low in mammals of 3%). In light of these results, Buckner et al. make the case for why voucher specimens need to be retained for every genome and provide a best-practices workflow that researchers should follow for obtaining, using and referencing samples used in genomic analyses.

This is a well-written and timely article that addresses a topic that has received some discussion in museum circles, but not often in the world of genomics. Some genomics sequencing efforts are highly cognizant of the issue and are making vouchers a priority (e.g., B10k), but others are less stringent in their requirements, so this should be a useful paper for impelling the use of vouchers that will educate genomic biologist about the needs for proper vouchering.

I also agree that when a high coverage genome is generated it is important for the data associated with that specimen to be curated in a single location. The authors make further very valid points regarding the importance of collection permits and the inclusion of local biologists.

Essential Revisions:

1. I would recommend expanding the references to various publications discussing the problems with the lack of preservation of voucher specimens for verification of taxonomic identification, determining correct taxonomic assignment of data with systematic revisions, and replication of studies (such as phylogenetic studies; Peterson et al. 2007; Pleijel et al. 2008). In addition to providing a broader view of the trend of the lack of voucher specimen being collected and preserved in many studies, the authors should address the following problematic areas:

2. The discussion on page 2 about the reasons why vouchers are important is clear and comprehensive but it would good to clarify what is and isn't acceptable as a "voucher". Is a cryobanked DNA sample adequate? A digital photograph? Or should there be some archive for physical photographs? While photographs may be useful as voucher specimens for most vertebrate groups (fish, amphibians, reptiles, birds) they are not particularly useful in other groups (small mammals such as rodents, shrews, and bats). Are there any publication that have examined the utility and limitations of photographs for species identification? Perhaps the Monk and Baker (2001) paper on e-vouchers would be informative. The authors should present a more critical evaluation of the utility of photographs and discuss further the limitations of even high-quality photos. Citations of "guidelines" for photographic vouchers would be helpful information. Steinke et al. (2009) provides information on obtaining high-quality digital photos of fish.

3. The authors should provide a more holistic view of a voucher specimen (see Kageyama et al. 2007; Cook et al. 2017; Lendemer et al. 2020) and include discussion of importance of vouchering duplicate tissue samples that will allow replication or validation of the genome assemblage as techniques improve.

4. The limitations described are mostly clear and correct. One thing that should be added is that in some cases, species are not just "rare" but are by law considered endangered and not allowed to be collected, and in some cases, not even handled. In these cases, such legalities should override the need for a voucher, or only a small invasive sample (blood, hair, feather, scale), or non-invasive sample may be available for collection. In these cases, some part of the original sample and the remnant DNA, if possible, should be considered the voucher, along with (again, if possible), detailed photographs or scans of the living organism (while in the hand ideally).

5. Also, as you get multiple genomes for a species or even a single population of a species, do you need to have vouchers for all of them? In some cases, permits will not be issued to collect more than one individual (or even one), so the remainder are obtained from blood samples. If the blood or DNA sample is archived and considered a voucher (perhaps along with one or more photographs), then this is reasonable. But otherwise, it would become difficult and cumbersome (especially since more and more studies are now involving dozens to hundreds of genomes for a single species).

6. I think it is admirable to note the issues of legality of collection and involvement and recognition of historically marginalized groups in genome research, but perhaps a better linkage to the question of vouchers for genomes could be made?

7. The idea of "proxy" specimens collected at the same locality and collection event is a particularly good suggestion for situations where the entire specimen is depleted during sequencing. I would suggest that the authors consider referring to these as 'paravoucher' specimens and genseq-6 category (following the nomenclature of Chakrabarty et al. 2013).

8. Organellar DNA data collected during the genomic sequencing process may not be a reliable method of taxonomic verification. It has been shown though that cytochrome b sequences from mitogenomes assembled from anchored hybrid data collection can demonstrate levels of divergence on the order of 9% comparable to Cytb sequences obtained by Sanger sequencing from the same taxon. It is still unclear how much noise may be present in the mitogenome assemblages that could prevent reliable taxon verification. To be a useful voucher, the barcode marker should be assembled, compared with reference sequences, accessioned into GenBank and linked by the Accession number to the genome concurrently with the release of the genomic assemblage.

However, DNA barcodes (Hebert et al. 2003; Rubinoff 2006; Waugh 2007; Krishnamurthy and Francis 2012; Pecnikar and Buzab 2014) do warrant discussion as a method of specimen verification. Using non-invasive methods for sample collection makes this approach especially important for vouchering rare species. For small mammals, a cytochrome b sequence is more reliable than photographs for species identification. Sequences of mitochondrial markers obtained by Sanger sequencing, entered in GenBank, and linked to the genome sequence by it GenBank Accession number is an alternative vouchering method that should be considered and discussed.

9. The text and Figure 2 omit an important aspect of vouchering, the curation of the voucher specimen. Several papers including Bradley et al. (2014) and Gropp (2020) have addressed the need for additional support (funding) for Natural History Collections. The authors should point to this need and encourage that funding of genomic studies to include some support for the long-term curation of the voucher specimens and associated tissue samples that will allow verification of species identification and replication of the genomic study.

10. How practical is the whole voucher approach? Buckner et al. note that very few vouchers have been deposited from whole genomes, but I wonder if this is exclusion or just circumstance. T sequence a whole genome, high concentrations of DNA need to be extracted. This is typically obtained from a living individual, from an immediate post mortem, or from a sample that had been frozen immediately post mortem. As Buckner et al. note, if a species is small then the entire sample will be destroyed in the process, or if a species is large, then more often it is simply a blood/tissue sample taken from a living individual (either from animals in captivity or from wild ones). These cases would therefore not leave a specimen available for accessioning into a museum collection. There is also the issue with post-mortem samples, where in order to obtain the DNA the specimen is left incomplete /in a non-pristine condition. Would these partial samples be practical for meaningful downstream morphological investigation, and would museums be willing to curate these partial remains?

11. Clearly a hard-line policy on complete voucher specimens for all whole genomes would be impractical, but that doesn't mean that it shouldn't be encouraged. A dialogue with museums on the data they would be willing to store would be timely. There is a clear need for data associated with genomes to be curated, some photographs and where possible the actual specimen, making museums the ideal candidates for this role. Also, just because the infrastructure might not currently be in place to deal with large scale voucher specimen deposits doesn't mean that it shouldn't be recognised as important. Investments could be considered to facilitate this.

References:

Bradley, R. D., L. C. Bradley, H. J. Garner, and R. J. Baker. 2014. Assessing the value of natural history collections and addressing issues regarding long-term growth and care. BioScience 64:1150-1158.

Cook, J. A., K. E. Galbreath, K. C. Bell, M. L. Campbell, S. Carriere, J. P. Colella, N. G. Dawson, J. L. Dunnum, R. P. Eckerlin, V. Fedorov, S. E. Greiman, G. M. S. Haas, V. Haukisalmi, H. Henttonen, A. G. Hope, D. Jackson, T. S. Jung, A. V. Koehler, J. M. Kinsella, D. Krejsa, S. J. Kutz, S. Liphardt, S. O. MacDonald, J. L. Malaney, A. Makarikov, J. Martin, B. S. McLean, R. Mulders, B. Nyamsuren, S. L. Talbot, V. V. Tkach, A. Tsvetkova, H. M. Toman, E. C. Waltari, S. Whitman, and E. P. Hoberg. 2017. The Beringian coevolution project: holistic collection of mammals and associated parasites reveal novel perspectives on evolutionary and environmental changes in the north. Arctic Science 3:585-617.

Gropp, R. E. 2020. Natural history collections are required to advance science, solve problems. BioScience 70:943.

Hebert, P. D. N., A. Cywinska, S. L. Ball, and J. R. deWaard. 2003. Biological identifications through DNA barcodes. Proceedings of the Royal Society B: Biological Science 270:313-321.

Kageyama, M., R. R. Monk, R. D. Bradley, G. F. Edson, and R. J. Baker. 2007. The changing significance and definition of the biological voucher. Pp. 257-264, in Museum Studies: Perspectives and Innovations (S. L. Williams and C. A. Hawks, eds.). Society for the Preservation of Natural History Collections, Yale University, New Haven, CT.

Krishnamurthy, P. K., and R. A. Francis. 2012. Autility of DNA barcoding in biodiversity conservation. Biodiversity and Conservation 21:1901-1919.

Lendemer, J., B. Thiers, A. K. Monfils, J. Zaspel, E. R. Ellwood, A. Bentley, K. Levan, J. Bates, D. Jennings, D. Contreras, L. Lagomarsino, P. Mabee, L. S. Ford, R. Guralnick, R. E/ Gropp, M. Revelez, N. Cobb, K. Seltmann, and M. C. Aime. 2020. The extended specimen network: a strategy to enhance US biodiversity collections, promote research and education. Bioscience 70:23-30.

Monk, R. R., and R. J. Baker. 2001. e-Vouchers and the use of digital imagery in natural history collections. Museology 10:1-8.

Pecnikar, Z, F., and E. V. Buzan. 2014. 20 years since the introduction of DNA barcoding: from theory to application. Journal of Applied Genetics 55:43-52.

Peterson, A. T., R. G. Moyle, A. S. Nyari, M. B, Robbins, R. T. Brumfield, and J. V. Remsen. 2007. The need for proper vouchering in phylogenetic studies of birds. Molecular Phylogenetics and Evolution 45:1042-1044.

Pleijel, F., U. Jondelius, E. Norlinder, A. Nygren, B. Oxelman, C, Schander, P. Sundberg, and M. Thollesson. 2008. Phylogenies without roots? A plea for the use of vouchers in molecular phylogenetic studies. Molecular Phylogenetics and Evolution 48:369-371.

Rubinoff, D. 2006. Utility of mitochondrial DNA barcodes in species conservation. Conservation Biology 20:1026-1033.

Steinke, D., R. Hanner, and P. D. Hebert. 2008. Rapid high-quality imaging of fish using a flat-bed scanner. Ichthyologial Research 56:210-211.

Waugh, J. 2007. DNA barcoding in animal species: progress, potential and pitfalls. Bioessays 29:188-197.

---

## [Author Response]

Essential Revisions:1. I would recommend expanding the references to various publications discussing the problems with the lack of preservation of voucher specimens for verification of taxonomic identification, determining correct taxonomic assignment of data with systematic revisions, and replication of studies (such as phylogenetic studies; Peterson et al. 2007; Pleijel et al. 2008). In addition to providing a broader view of the trend of the lack of voucher specimen being collected and preserved in many studies, the authors should address the following problematic areas:

We agree and have added the suggested papers to the manuscript and reference list among others. Importantly, the inclusion of Peterson et al. 2007 provides additional support for our main argument: “we do not believe that such unvouchered studies fit the definition of ‘science.’’; study cannot be replicated, the museum community, in particular, has thrived on a tradition of open exchange of material among researchers and institutions”. We have also included Pleijel et al. 2008 in several locations throughout the manuscript (see also #7, below).

2. The discussion on page 2 about the reasons why vouchers are important is clear and comprehensive but it would good to clarify what is and isn't acceptable as a "voucher". Is a cryobanked DNA sample adequate? A digital photograph? Or should there be some archive for physical photographs? While photographs may be useful as voucher specimens for most vertebrate groups (fish, amphibians, reptiles, birds) they are not particularly useful in other groups (small mammals such as rodents, shrews, and bats). Are there any publication that have examined the utility and limitations of photographs for species identification? Perhaps the Monk and Baker (2001) paper on e-vouchers would be informative. The authors should present a more critical evaluation of the utility of photographs and discuss further the limitations of even high-quality photos. Citations of "guidelines" for photographic vouchers would be helpful information. Steinke et al. (2009) provides information on obtaining high-quality digital photos of fish.

We agree with the referees that photographs are useful, but limited as a stand-in for physical specimens. To try and encapsulate some of the points made above, we have added citations for readers linking them to a manuscript about the limits of photographic evidence (Ceríaco, L.M., Gutiérrez, E.E., Dubois, A. and Carr, M., 2016. Photography-based taxonomy is inadequate, unnecessary, and potentially harmful for biological sciences. Zootaxa, *4196*(3), pp.435-445), and we also now cite Monk and Baker’s paper on e-vouchering (see #3) in place of Steinke et al. In particular, Steinke et al. is rather specific to fishes and rapid photography, while Monk and Baker is more general and appears to cover many of the same useful points regarding the utility of photographic images for specimens rather than limits. To attempt to summarize argumentation on both the value and the costs associated with photographic vouchers, we have updated the revision to read: “In these cases, detailed photographs should be taken to aid future identification attempts although it should also be recognized that photographs have limited utility for taxonomy.^11,12^ Many species are distinguished on the basis of inconspicuous characters or internal anatomy that photographs might not capture.”

We also discuss what we consider “secondary” vouchers like the kinds mentioned above (see #3, below).

3. The authors should provide a more holistic view of a voucher specimen (see Kageyama et al. 2007; Cook et al. 2017; Lendemer et al. 2020) and include discussion of importance of vouchering duplicate tissue samples that will allow replication or validation of the genome assemblage as techniques improve.

We have added this more holistic description about voucher specimens, however, we do not think just having additional tissue samples is enough. Thank you for recommending Kageyama et al. 2007 – which we use to support the accessory material explanation because of the “secondary voucher” description in that paper, however we think it is confusing to dilute the term “voucher” in our paper from the definition we present in the first sentence and that is often used elsewhere. To clarify we added the following lines:

“These types of accessory or partial biological samples and photographs (or other so-called *e*-vouchers^15^) fall in the category of “secondary vouchers”^16^ that should be used in support of vouchering whole specimens, not as alternatives – unless collecting a specimen is not possible. This holistic approach to vouchering, where primary and secondary voucher materials are collected and stored together will further increase the repeatability and reliability of genomic studies.”

The Cook et al. reference, is a good one for explaining parasites and other materials that may be associated with a voucher but is perhaps not appropriate for this DNA-focused discussion above so we have added it as part of a discussion of the other uses of vouchers. Lendemer et al. is already cited in the manuscript and although the ‘extended specimen’ view is a very important one we focused here again on the DNA voucher aspects rather than move into a discussion of why vouchers are important for many kinds of studies. However we cite both of these in a line where we explain: “Vouchering also enables a wide spectrum of scientific uses beyond genomics including additional studies of natural history and ecology and the use of specimen resources for outreach activities^3,24^.” We think this and other sections help expand the holistic view of vouchers that was recommended by the reviewers and editors.

4. The limitations described are mostly clear and correct. One thing that should be added is that in some cases, species are not just "rare" but are by law considered endangered and not allowed to be collected, and in some cases, not even handled. In these cases, such legalities should override the need for a voucher, or only a small invasive sample (blood, hair, feather, scale), or non-invasive sample may be available for collection. In these cases, some part of the original sample and the remnant DNA, if possible, should be considered the voucher, along with (again, if possible), detailed photographs or scans of the living organism (while in the hand ideally).

That is a fair point and we have added discussion of these limitations by adding “too rare to be legally obtained” in the 2^nd^ sentence of the ‘Limitations section’ – of course there is no mechanism for collecting genomic DNA from an organism that cannot be ‘handled’ as you say above, but we think the additional section about “secondary vouchers” illustrate the other mechanisms for tactics that can be used when a voucher cannot be obtained. And we also have a section about the importance of vouchers to establish legality of collections in the “Improving Legality, Equity, and Inclusion in Genomics” section.

5. Also, as you get multiple genomes for a species or even a single population of a species, do you need to have vouchers for all of them? In some cases, permits will not be issued to collect more than one individual (or even one), so the remainder are obtained from blood samples. If the blood or DNA sample is archived and considered a voucher (perhaps along with one or more photographs), then this is reasonable. But otherwise, it would become difficult and cumbersome (especially since more and more studies are now involving dozens to hundreds of genomes for a single species).

Even in the hypothetical described above we think vouchers should be part of a best practice. However, as we mention in the proxy voucher section, one member of a “lot” can suffice as a representative (see #7 below). We now add as part of our last point in the best practices section *“As we enter a future when genomics will be the most frequent form of genetic study, we need to avoid a scenario where it will become increasingly intractable to correctly assign species to available genome assemblies; having a voucher specimen representing the reference genome for every species is the best solution to that increasingly difficult problem.”* We hope the emphasis on every *species* will show that a voucher may not be needed for every individual sampled in the future.

6. I think it is admirable to note the issues of legality of collection and involvement and recognition of historically marginalized groups in genome research, but perhaps a better linkage to the question of vouchers for genomes could be made?

We have made this effort because we think it is an important point and added to our argument. Line 35-46 now read: “Minimally, vouchering ensures the record of the collectors who enable these studies is preserved: the names of original collectors are linked to the specimens and should be perpetuated with the data obtained from their vouchers. […] Vouchering also enables a wide spectrum of scientific uses beyond genomics including additional studies of natural history and ecology and the use of specimen resources for outreach activities^3,24^.”

7. The idea of "proxy" specimens collected at the same locality and collection event is a particularly good suggestion for situations where the entire specimen is depleted during sequencing. I would suggest that the authors consider referring to these as 'paravoucher' specimens and genseq-6 category (following the nomenclature of Chakrabarty et al. 2013).

We do like the suggested term “paravoucher” however, Pleijel et al. 2008 (recommended in #1 above) suggest an equivalent term which already exists “paragenophore” and we cite this in our text. From Pleijel et al., 2008: “A paragenophore is an individual organism collected at the same time and place as the study organism, and identified by the author as belonging to the same operational taxonomic unit. The voucher in this case is another individual than the one used for the molecular study; however, since it is collected at the same time and from the same locality, it is deemed likely to belong to the same population”. Because this other term exists we remove the reference to GenSeq-6 in order to decrease the amount of jargon from other papers (even though it is one of my own).

8. Organellar DNA data collected during the genomic sequencing process may not be a reliable method of taxonomic verification. It has been shown though that cytochrome b sequences from mitogenomes assembled from anchored hybrid data collection can demonstrate levels of divergence on the order of 9% comparable to Cytb sequences obtained by Sanger sequencing from the same taxon. It is still unclear how much noise may be present in the mitogenome assemblages that could prevent reliable taxon verification. To be a useful voucher, the barcode marker should be assembled, compared with reference sequences, accessioned into GenBank and linked by the Accession number to the genome concurrently with the release of the genomic assemblage.

That is an interesting point and suggestion. We don’t think we can mention the above statistic (mitogenome from genomics versus Sanger barcoding) without a citation.

However, DNA barcodes (Hebert et al. 2003; Rubinoff 2006; Waugh 2007; Krishnamurthy and Francis 2012; Pecnikar and Buzab 2014) do warrant discussion as a method of specimen verification. Using non-invasive methods for sample collection makes this approach especially important for vouchering rare species. For small mammals, a cytochrome b sequence is more reliable than photographs for species identification. Sequences of mitochondrial markers obtained by Sanger sequencing, entered in GenBank, and linked to the genome sequence by it GenBank Accession number is an alternative vouchering method that should be considered and discussed.

Unfortunately, the Barcode of Life project is even more fraught with misidentifications (and voucher-less sequences) than GenBank (Meier et al., 2008 https://academic.oup.com/sysbio/article/57/5/809/1619912?login=true). Barcodes are only as good as the identifications associated with the source organism. We now cite this recent study that illustrates the problem with simple Barcode IDs:

Pentinsaari, M., Ratnasingham, S., Miller, S. E., and Hebert, P. D. (2020). BOLD and GenBank revisited–Do identification errors arise in the lab or in the sequence libraries?. PloS one, *15*(4), e0231814.

We also now explicitly mention the Barcode of Life, along with GenBank as follows. “Alternatively, comparative organellar DNA can also be misidentified or unavailable from public databases such as the Barcode of Life and GenBank^18^.” And we reference ‘barcode genes’ elsewhere to illustrate our point about how they cannot be used with confidence in lieu of a voucher.

9. The text and Figure 2 omit an important aspect of vouchering, the curation of the voucher specimen. Several papers including Bradley et al. (2014) and Gropp (2020) have addressed the need for additional support (funding) for Natural History Collections. The authors should point to this need and encourage that funding of genomic studies to include some support for the long-term curation of the voucher specimens and associated tissue samples that will allow verification of species identification and replication of the genomic study.

Yes additional funding would help. We now cite the Bradley et al. article in this sentence: “Support (financial as well as academic credit) for museums and preparators who maintain these research collections and update taxonomy and reference catalogs should also not be overlooked^2^”. We don’t expand on this topic too much here because of a lack of space and it may take us away too much from our point about recognizing collectors.

10. How practical is the whole voucher approach? Buckner et al. note that very few vouchers have been deposited from whole genomes, but I wonder if this is exclusion or just circumstance. T sequence a whole genome, high concentrations of DNA need to be extracted. This is typically obtained from a living individual, from an immediate post mortem, or from a sample that had been frozen immediately post mortem. As Buckner et al. note, if a species is small then the entire sample will be destroyed in the process, or if a species is large, then more often it is simply a blood/tissue sample taken from a living individual (either from animals in captivity or from wild ones). These cases would therefore not leave a specimen available for accessioning into a museum collection. There is also the issue with post-mortem samples, where in order to obtain the DNA the specimen is left incomplete /in a non-pristine condition. Would these partial samples be practical for meaningful downstream morphological investigation, and would museums be willing to curate these partial remains?

Museums do keep partial remains, and we have found ourselves in this same scenario where a large portion of a specimen is depleted. Luckily, non-pristine specimens, necropsied individuals, and other elements can be deposited in museum collections. We have now modified this section with the following lines: “Live organisms can be treated as vouchers and can be provided museum catalog numbers to ensure future preservation upon their death; even if a specimen is heavily dissected from a necropsy, many permanent collections are willing to preserve partial remains as vouchers. Similarly, samples collected from organisms that are extremely large (such as blood/tissue samples taken from a whale) can also be curated and stored with other biological sample data in most natural history collections.”

11. Clearly a hard-line policy on complete voucher specimens for all whole genomes would be impractical, but that doesn't mean that it shouldn't be encouraged. A dialogue with museums on the data they would be willing to store would be timely. There is a clear need for data associated with genomes to be curated, some photographs and where possible the actual specimen, making museums the ideal candidates for this role. Also, just because the infrastructure might not currently be in place to deal with large scale voucher specimen deposits doesn't mean that it shouldn't be recognised as important. Investments could be considered to facilitate this.

We completely agree, and have throughout the revision highlighted the need to support museums and their critical role in the vouchering process. We also soften the ‘hard-line’ of always vouchering with our substantial section about Limitations because we recognize that some situations will never fit with a general recommendation. Thank you for helping us with the expanded view based on the comments above. We think this revised take will allow for the many kinds of collections being made today for genomic research.

Thank you for the list of papers. We have incorporated them into the paper, except a few that are mostly restricted to discussing barcode genes or are too limited in their scope. We also didn’t want to go too well beyond the 20 reference limit for this kind of submission.